# Bad Smells and Broken DNA: A Tale of Sulfur-Nucleic Acid Cooperation

**DOI:** 10.3390/antiox10111820

**Published:** 2021-11-17

**Authors:** Rodney E. Shackelford, Yan Li, Ghali E. Ghali, Christopher G. Kevil

**Affiliations:** 1Department of Pathology and Translational Pathobiology, Louisiana State University Health Sciences Center, Shreveport, LA 71130, USA; yan.li@lsuhs.edu (Y.L.); chris.kevil@lsuhs.edu (C.G.K.); 2Head & Neck Oncologic/Microvascular Reconstructive Surgery Department of Oral & Maxillofacial/Head & Neck Surgery, Louisiana State University Health Sciences Center, Shreveport, LA 71130, USA; gghali@lsuhs.edu

**Keywords:** hydrogen sulfide, DNA repair, cystathionine β-synthase, cystathionine γ-lyase, 3-mercaptopyruvate sulfurtransferase, ATR, MEK1, autophagy

## Abstract

Hydrogen sulfide (H_2_S) is a gasotransmitter that exerts numerous physiologic and pathophysiologic effects. Recently, a role for H_2_S in DNA repair has been identified, where H_2_S modulates cell cycle checkpoint responses, the DNA damage response (DDR), and mitochondrial and nuclear genomic stability. In addition, several DNA repair proteins modulate cellular H_2_S concentrations and cellular sulfur metabolism and, in turn, are regulated by cellular H_2_S concentrations. Many DDR proteins are now pharmacologically inhibited in targeted cancer therapies. As H_2_S and the enzymes that synthesize it are increased in many human malignancies, it is likely that H_2_S synthesis inhibition by these therapies is an underappreciated aspect of these cancer treatments. Moreover, both H_2_S and DDR protein activities in cancer and cardiovascular diseases are becoming increasingly apparent, implicating a DDR–H_2_S signaling axis in these pathophysiologic processes. Taken together, H_2_S and DNA repair likely play a central and presently poorly understood role in both normal cellular function and a wide array of human pathophysiologic processes. Here, we review the role of H_2_S in DNA repair.

## 1. Introduction

The maintenance of genomic stability is essential for life, and cells have evolved complex and intricate molecular machinery to ensure DNA stability and accurate DNA replication [1,2,3,4]. Eukaryotic cells carry two separate genomes with different evolutionary origins [4,5]. The nuclear genome is diploid, linear, and in humans contains roughly 3.3 billion base pairs encoding over 20,000 genes [4,5]. Conversely, the mitochondrial genome is circular, contains 37 genes in 16,569 base pairs, and occurs in multiple copies at 100–1000/cell [4,5]. The two genomes extensively interact, with the nuclear genome encoding roughly 1500 mitochondrial proteins, including those involved in mitochondrial DNA repair, while mitochondrial genomic damage can initiate apoptotic cell death via cytochrome c release and can also activate the innate immune response [4,5,6,7]. Hydrogen sulfide (H_2_S) is a gasotransmitter that, along with nitric oxide and carbon monoxide, functions in a vast number of different physiologic and pathophysiologic processes [8,9]. Specifically, H_2_S has many physiologic regulatory roles, including in the renal, cardiovascular, central nervous, and digestive systems, and is also dysregulated in many different pathologic processes including cancer, cardiovascular diseases, and neurodegeneration [10,11,12,13,14,15,16,17]. Recently, H_2_S has been found to regulate mitochondrial and nuclear DNA stability and repair [11,12,13]. Here, we review this new area of inquiry and discuss its possible implications for cancer chemotherapy and cardiovascular diseases.

## 2. H_2_S Chemistry, Synthesis, and Catabolism

### 2.1. H_2_S Chemistry

H_2_S has been known for over 300 years as an environmental toxin with high H_2_S concentrations causing damage in multiple organs, difficulty with breathing, shock, and convulsions, which may lead to death [18]. H_2_S toxicity occurs following cytochrome c oxidase, carbonic anhydrase, Na^+^/K^+^ ATPase, monoamine oxidase, and possibly ATR kinase inhibition [12,18,19,20,21]. H_2_S is a colorless, weak diprotic acid with a characteristic rotten egg smell, with first and second pKa values of 6.76 and 19 at 37 °C [18]. At pH 7.4, H_2_S is ~70–80% HS^−^, ~20% H_2_S, with low very concentrations of S_2_^−^ [19]. H_2_S readily diffuses across biological membranes [22]. HS^−^ has high nucleophilicity and chemical reactivity, but it is not membrane permeable, although it may cross membranes due to its rapid interconversion with H_2_S or through specific transporters [9,22,23,24]. The minus two sulfur oxidation state in H_2_S renders it an obligatory reductant, and H_2_S exerts numerous antioxidant effects [9,24,25,26]. However, cellular H_2_S concentrations are low at 10–30 nM, and H_2_S reacts too slowly with oxidants, such as H_2_O_2_ and hypochlorite, to exert significant antioxidant effects [14,19,27,28,29]. Thus, its antioxidants effects are likely mediated through events such as the H_2_S-mediated induction of glutathione synthesis and cystine uptake, the inhibition of mitochondrial free radical production, and Nrf2 induction [9,25,27]. H_2_S exists in thermodynamic equilibrium with other sulfur species including persulfides, polysulfides, and reactive sulfur species. The roles of these compounds in human health and disease are presently poorly understood and are an area of intense investigation [14]. The cell and tissue H_2_S half-lives are short, being only a few minutes [9,19].

### 2.2. H_2_S Synthesis

Biological H_2_S synthesis was first identified in bacteria in 1895, and it was not until 1996 that endogenous H_2_S as a biological modulator in humans was identified [10,30]. In mammals H_2_S is primarily synthesized by three systems which contribute unequally to the sulfur pool: (1) enzymatic synthesis, (2) non-enzymatic synthesis, and (3) microbiome production. Enzymatic H_2_S synthesis occurs through three enzymes: cystathionine γ-lyase (CSE), cystathionine β-synthase (CBS), and 3-mercaptopyruvate sulfurtransferase (3-MST, [9]). These enzymes show tissue and organ-specific distributions patterns [8,10,31,32]. CSE is predominantly found within the vasculature, while CBS occurs in the liver, brain, and nervous system, and 3-MST occurs mainly in the vasculature and the brain [10,31,32]. CSE and CBS are predominantly cytosolic pyridoxal-5′-phosphate-dependent hemeproteins, while 3-MST is mainly mitochondrial and synthesizes roughly 90% of brain H_2_S [10,31,32,33].

The enzymatic synthesis of H_2_S in mammals requires methionine and cysteine, with methionine being derived solely from the diet, as it cannot be synthesized in mammalian tissues [19,34]. These amino acids are converted into homocysteine by the transsulfuration pathway [19,34]. CBS synthesizes H_2_S by catalyzing homocysteine and L-cysteine to form cystathionine and H_2_S [19,35]. CBS generates H_2_S through a β-replacement reaction also producing serine. In the presence of homocysteine H_2_S synthesis increases 23-fold compared to the reaction with L-cysteine alone [19,36]. CBS also catalyzes the condensation of L-serine and homocysteine, forming cystathionine and H_2_O, a significant step in L-cysteine biosynthesis [37]. CSE catalyzes homocysteine, generating H_2_S, α-ketobutyrate, and NH_3_ [9,19,38]. Interestingly, at physiologic L-cysteine and homocysteine concentrations, roughly 70% of the H_2_S synthesized for the CSE-mediated α,β-elimination of L-cysteine, while the α,γ-elimination of homocysteine contributes approximately 29% of the total H_2_S content; however, roughly 90% of the H_2_S is derived from the α,γ-elimination of homocysteine when its levels are increased to those seen in hyperhomocysteinemia [38]. 3-MST H_2_S synthesis requires cysteine aminotransferase to convert L-cysteine into 3-mercaptopyruvate, which is then catalyzed by 3-MST into H_2_S and pyruvate [39]. D-cysteine can be converted to 3-mercaptopyruvate by the peroxisome-located D-amino acid oxidase, which, once imported into the mitochondrion, is converted into H_2_S by 3-MST. As D-amino acid oxidase expression is limited to the brain and kidneys, this H_2_S synthesis pathway is likely limited to those organs [40].

Based on its chemistry, the molecular mechanisms of H_2_S reactivity have been placed into three categories: (1) chemical interfacing/scavenging with reactive oxygen and nitrogen species, (2) chemical modification of protein cysteines to persulfides, and (3) binding to and/or redox reactions with metal centers [9]. Intracellular H_2_S exists as free H_2_S, acid-labile sulfide, and bound sulfane sulfur [19]. The acid-labile sulfide faction consists of sulfur present in iron–sulfur clusters contained within iron–sulfur proteins (non-heme), which are ubiquitous and include ferredoxins, rubredoxins, aconitase, and succinate dehydrogenase [19]. The sulfane sulfur fraction consists of sulfur atoms bound only to other sulfur atoms, these include thiosulfate S_2_O_3_^2–^, persulfides, R–S–SH, thiosulfonates R–S(O)–S–R’, polysulfides R–S_n_–R, polythionates S_n_O_6_^2–^, and elemental sulfur S^0^ [19]. Free cellular H_2_S represents less than 1% of the potentially available sulfide, indicating that the endogenous sulfide pool likely has significant buffering capacity [9].

### 2.3. Non-Enzymatic H_2_S Synthesis

H_2_S is also synthesized through non-enzymatic mechanisms and occurs by an iron and vitamin B_6_-mediated catalysis of L- or D-cysteine producing pyruvate, NH_3_, and H_2_S [41]. This H_2_S synthesizing pathway likely plays a role in maintaining basal H_2_S levels and may be an important H_2_S source in iron overload and hemorrhagic disorders [41]. Lastly, the mammalian microbiome regulates systemic H_2_S bioavailability and metabolism. Germ-free mice show significantly lower plasma and gastrointestinal H_2_S and 50–80% lower plasma, adipose, and lung tissue bound sulfane sulfur compared to conventionally housed mice. Interestingly, CSE activity was reduced in many organs of the germ-free mice, while tissue cysteine levels were elevated [42].

### 2.4. H_2_S Catabolism

High cellular H_2_S concentrations can be toxic, and excess H_2_S is predominately removed by the stringently regulated mitochondrial sulfur oxidation pathway [43]. H_2_S catabolism is initiated by the mitochondrial matrix flavoprotein sulfide quinone oxidoreductase (SQR), which oxidizes H_2_S to form an SQR-persulfide intermediate. The persulfide is then transferred to glutathione to form glutathione persulfide, which is further oxidized by the ethylmalonic encephalopathy 1 or thiosulfate sulfurtransferase proteins to form sulfite or thiosulfate, respectively [43,44,45]. The sulfite and thiosulfate are ultimately excreted in the urine [46]. The electrons released by SQR enter complex III of the electron transport chain and are used to generate ATP, making H_2_S an inorganic compound capable driving mitochondrial ATP synthesis [47]. Outside of this review, however, polysulfides can be synthesized by all three H_2_S-sythesizing enzymes and increasingly have been found to play important functions in many physiologic and pathophysiologic processes [29,48,49].

## 3. Life’s Origin and H_2_S

### 3.1. Life’s Origin and H_2_S

Biochemical, fossil, and molecular clock dating methods indicate that life first appeared 3.7 to 4.2 billion years ago in a reducing, ferruginous, and euxinic environment probably at hydrothermal vents rich in NH_3_, N_2_, CO_2_, CO, CH_4_, H_2_, H_2_S, and dissolved metals, especially Fe^2+^ and Mn^2+^ [50,51,52,53,54,55]. Due to the fact of its ubiquity on the early Earth and versatile chemistry, H_2_S likely played an essential role in prebiotic chemistry and the emergence of life [52,54,55]. Support for this comes from analyses demonstrating that reactive oxygen and reactive sulfur species detoxifying mechanisms have been present since the origin of life, some four billion years ago, and have continued to the present in the biochemistry of the Archaea, Bacteria, and Eukarya [50,51,52,53,54,55,56]. Since genomic stability maintenance is an ancient and absolute requirement for life, and H_2_S biochemistry is similarly ancient and ubiquitous, it is very likely that H_2_S functioned in the earliest biochemical pathways including in those regulating genomic stability [1,2,3,11,12,13,17,52,53,54,55,56,57,58].

### 3.2. H_2_S and the DNA Damage Response

The DNA damage response (DDR) comprises a complex network of cellular pathways that cooperatively detect DNA damage, signals its presence, and promotes DNA repair, maintaining genomic stability [1,2,3]. Phylogenomic analyses indicate that many elements of the Eukaryotic DDR are ancient, appearing in the first Metazoa and subsequently undergoing evolutionary diversification [59].

Extensive evidence indicates that H_2_S affects cell DNA stability, impinging on the DDR and cell viability. For example, when the nuclei from Chinese hamster ovary cells were treated 2 h with 1 μM Na_2_S, they exhibited significant DNA damage that was attenuated by treatment with the antioxidant butyl-hydroxyanisole [60]. Treatment of two glioblastoma cell lines with a high amount of Na_2_S (476 μM) for 4 h, increased DNA damage, oxidative stress levels, and increased γ-H2AX foci formation [61]. Additionally, treatment of human intestinal epithelial cells with very high (1–2 mM) Na_2_S induced DNA breaks as measured by the comet assay [62]. Similarly, human lung fibroblasts treated 12 h with 10 μM NaHS showed micronuclei formation, increased p21, p53, Bax, cytochrome c, Ku-70 and Ku-80 expression, and a G_1_ checkpoint response [63]. These studies implicate H_2_S in the DDR as; (1) broken DNA activates the DDR, (2) γ-H2AX foci formation requires the activities of the DDR proteins ATM, ATR, and DNA-PK, (3) the Ku70/Ku80 heterodimer associates with DNA-PK to promote DNA repair, and (4) the ATM kinase is required for the oxidative stress-induced G_1_ checkpoint response and rapid p53 induction [1,2,3,64].

Other studies have shown that H_2_S can increase/preserve DNA stability. For example, in a murine model, a unilateral nephrectomy with contralateral ureteral obstruction suppressed H_2_S kidney levels and caused more DNA damage in CSE deficient mice compared to wild-type mice, indicating that CSE expression plays a role in maintaining DNA stability upon ischemia/reperfusion injury [65]. Additionally, daily injection of the H_2_S donor diallyl sulfide intraperitoneally into female rats at 50 mg/kg induced p53, Gadd45a, PCNA, and DNA polymerase δ in their breast tissue, suggesting that H_2_S enhances breast tissue DNA repair capacity [66]. Interestingly, exogenous H_2_S also affects the mitochondrial genome. For example, CSE deficient murine smooth muscle and aortic tissue showed reduced mitochondrial DNA copy numbers, mitochondrial content, mitochondrial-specific mRNAs (*MT-CO1*, *CytB*, and *Atp 6*), and mitochondrial transcription factor A (TFAM) mRNA and protein expression, and it elevated DNA methyltransferase 3A (Dnmt3a) expression, accompanied by increased global DNA methylation with increased TFAM promoter methylation [67]. Treatment with 30 and 60 μM NaSH for 48 h reversed these effects, with increased mitochondrial marker expression (mitochondrial DNA copy numbers, mRNAs, and mitochondrial content) and decreased Dnmt3a and TFAM promoter methylation, increasing TFAM expression [67]. This study indicates that CSE-derived H_2_S plays an important role in the maintenance of mitochondrial function and genomic stability. Lastly, 30–100 nM concentrations of AP39, a mitochondria-targeted H_2_S donor, increased endothelial cell H_2_S levels and stimulated mitochondrial electron transport and bioenergetic functions. Treatment of the endothelial cells with oxidative stress increased reactive oxygen species (ROS), reduced cell viability, suppressed cellular bioenergetics, and increased mitochondrial DNA damage, events reversed by 100 nM AP39 treatment [68]. Taken together, these studies indicate that under different conditions H_2_S can increase DNA damage or suppress it and also impinges on the DDR. A caveat to keep in mind is that many experimental procedures in these studies used high concentrations of H_2_S donors that are likely non-physiologic [61,62]. Additionally, many studies of H_2_S biology use H_2_S synthesis inhibitors that have low specificity, complicating experimental result interpretation (reviewed in [69]).

## 4. H_2_S and Mitochondrial DNA Repair

The mitochondria are the major cellular site for ROS generation, and the mitochondrial genome is subject to significant DNA, protein, and lipid oxidative damage [70]. Mitochondrial DNA repair is distinct from and, in general, less complex than the nuclear DNA repair systems. For example, base-excision repair (BER) predominates, while nucleotide excision repair (NER) is absent [5,71]. Moreover, mitochondrial genomes with double-stranded DNA (dsDNA) breaks are usually rapidly degraded, leading to a drop in genome copy number, which are replaced through non-cleaved genome replication, often leading to a shift in heteroplasmy [5]. A role for H_2_S in mitochondrial function is well established with, for example, the mitochondrial H_2_S donor AP39 promoting mitochondrial bioenergetics and genomic stability and in the face of exogenous oxidants [68]. Additionally, in ovarian cancer cell lines, CBS expression maintains mitofusin-2 expression, with CBS knockdown lowering mitofusin-2 expression, causing mitochondrial fragmentation with a fused spherical morphology and increased unbranched mitochondria [72]. Mitofusin-2 exerts anti-apoptotic effects, and its ablation is lethal in mice [73]. Interestingly, its expression is lower in obesity, diabetes, and in animal models prone to atherosclerosis, and is increased by weight loss and exercise [73].

The apyrimidinic/apurinic endonuclease 1 (APE1), exonuclease G (EXOG), DNA Ligase III (LIG3), and DNA polymerase gamma (Pol γ) play central roles in mitochondrial BER [4,11,74,75,76]. Loss of these proteins has severe often lethal effects. For example, EXOG depletion induces persistent single-stranded DNA breaks leading to apoptosis, while APE1 ablation is embryonic lethal, and its removal by Cre expression causes apoptotic cell death within 24 h [74,75]. In the A549 lung adenocarcinoma cell line, siRNA knockdown of CBS, CSE, or 3-MST or treatment with the CSE-specific inhibitor D, L-propargylglycine (PAG) combined with exogenous oxidative stress significantly increased mitochondrial DNA damage [11]. Interestingly, the interactions of EXOG with APE1, LIG3, and POL were all attenuated with CBS, CSE, or 3-MST knockdown or pharmacologic CBS inhibition by aminooxyacetic acid (AOAA) [11]. The interactions of EXOG with APE1 or LIG3 following AOAA treatment were restored and mitochondrial DNA damage was reduced with AP39 co-treatment, demonstrating that mitochondrial H_2_S restored these interactions and increased mitochondrial genomic stability [11]. Mass spectrometric analysis revealed that EXOG Cys 76 was sulfhydrated, with the H_2_S donor NaHS increasing EXOC and APE1 interactions. Mutation of EXOG Cys 76 to alanine lowered its interactions with APE1 and made the interaction insensitive to NaHS treatment [11]. Thus, this elegant study demonstrated that mitochondrial H_2_S plays a central role in mitochondrial genomic stability and DNA repair.

## 5. H_2_S and Nuclear DNA Repair: ATR and MEK1

### 5.1. ATR

Nuclear DNA repair and the DDR involve at least five major pathways comprising BER, NER, mismatch excision repair, homologous recombination, and non-homologous end joining [1,2,3]. The *ataxia-telangiectasia mutated,* and RAD3-related serine/threonine protein kinase (ATR) plays a central role in the DDR, where it stabilizes single-stranded DNA (ssDNA) at stalled replication forks, lowers replication stress, initiates cell cycle checkpoints, and promotes faithful anaphase chromosomal segregation [1,2,3,12]. Interestingly, increased ATR/phospho-ATR expression is a poor prognostic factor in breast, bladder, and ovarian cancers [77,78,79]. Analysis of a colon adenocarcinoma cell lines with wild-type and biallelic knock-in hypomorphic ATR Seckel syndrome 1 genes revealed lower cellular H_2_S levels in the mutant cells compared to the wild type [12]. ATR inhibition with the pharmacologic ATR inhibitor NU6027 also significantly lowered cellular H_2_S levels in the wild-type but not the mutant cells [12]. Treatment of both cell lines with the CBS/CSE inhibitor β-cyano-L-alanine suppressed H_2_S levels in both cell types, demonstrating that cellular H_2_S levels are regulated by CBS/CSE and ATR, which form separate regulatory foci [12].

Interestingly, ATR activation correlates with serine 435 phosphorylation, an event also required for ATR-XPA dimer formation and subsequent NER [12,80]. Treatment of the colon adenocarcinoma cell lines with β-cyano-L-alanine increased this phosphorylation, while treatment with the H_2_S donor diallyl trisulfide significantly suppressed it in the wild-type but not mutant cells. UV light and oxidative stress treatments similarly induced this phosphorylation in the wild-type but not mutant cells [12]. Activated ATR phosphorylates the CHK1 kinase serine 345, leading to its activation [1,2,3,12]. When the cells lines were pretreated with β-cyano-L-alanine, followed by a low concentration of oxidative stress, oxidative stress-induced CHK1 phosphorylation increased with H_2_S synthesis inhibition, an event again not seen in the mutant cells [12]. Lastly, to examine the effects of these events on genomic stability, oxidative stress-induced dsDNA breaks were quantified in both cell types with and without H_2_S synthesis inhibition by β-cyano-L-alanine treatment. H_2_S synthesis inhibition caused low levels of oxidative stress to significantly induce dsDNA breaks, where otherwise they were not increased [12]. The mutant cells also showed increased breaks compare to the wild-type cells [12]. Taken together, these finding indicate that ATR regulates cellular H_2_S levels and H_2_S, in turn, regulates ATR phosphorylation, ATR kinase activity, and nuclear genomic stability [12].

### 5.2. MEK1

An important initial and required step in the DDR is carried out by poly (ADP-ribose) polymerases (PARPS) that transfer ADP-ribose from NAD^+^ to glutamic acid residues on a protein acceptor, creating ADP-ribose polymers at sights of DNA damage [1,2,3,81]. These chains function in the recruitment of factors involved in DNA repair such as polymerase β, XRCC1, and ligase IIIα [1,2,3,82]. PARP activation is tightly regulated by a cascade of kinases including the MEK/ERK signaling pathway [1,2,3,82,83,84]. In an interesting study, treatment of human endothelial cells for 2 h with 10 μM NaHS increased MEK1 Cys 341 S-sulfhydration. This event resulted in ERK1/2 phosphorylation and its subsequent translocation into the nucleus, where it activated PARP-1 through a direct interaction [13]. Mutation of MEK1 Cys 341 to Gly blocked these events.

Next HEK293 cells were treated with methyl methanesulfonate (MMS), which induces ssDNA and dsDNA breaks and activates PARP-1 activity, with PARP-1 then recruiting XRCC1 and DNA ligase III to initiate DNA repair [13,83,84,85]. MMS treatment of the HEK293 cells resulted in MEK1 Cys 341 S-sulfhydration, but not in cells carrying the Gly 341 mutated MEK1 [13]. This S-sulfhydration increased in CSE over-expressing HEK293 cells. Lastly, application of MMS to human endothelial cells, with and without co-treatment with 0.1–10 μM NaHS, increased PARP-1 activity in a dose-dependent manner, with PARP-1 activation detectable at 5 min with NaHS treatment and only at 30 min without NaHS treatment [13]. NaHS treatment also reduced the amount of HEK293 cell DNA damage and increased the amounts of XRCC1 and DNA ligase III recruited [13]. This study indicates that CSE-generated H_2_S acts as a DNA damage protectant, S-sulfhydrating MEK1 Cys 341, activating ERK1/2 and PARP-1 to repair DNA damage [13].

These studies on the roles of ATR and MEK1 in nuclear DNA repair demonstrate that H_2_S plays an important and, as yet, poorly defined role in nuclear DNA repair regulation [12,13]. Interestingly, low cellular H_2_S concentrations activate ATR, as measured by its kinase activity towards CHK1 [12]. Conversely, MEK1 activity in DNA repair is increased with increased H_2_S concentrations, as supplied either by exogenous NaHS or increased CSE expression [13]. Thus, these studies imply that nuclear DNA repair is likely modulated by both increased and decreased cellular H_2_S concentrations [12,13]. Additionally, CBS and CSE knockdown both attenuated the mitochondrial interactions of EXOG with APE1, LIG3, and POL γ, implying that ATR may also indirectly regulate mitochondrial BER [11,12].

## 6. H_2_S, Autophagy, and the DDR

Recently, Jiang et al. employed a high-content screen of ~12,000 with diversified chemical structures and molecular targets, screening for compounds that increased cellular H_2_S expression > 1.4 with a concomitant >50% cell survival compared to DMSO treated controls [85]. Interestingly, the largest activating compound class screened consisted of genotoxic compounds, with the most active compounds being the topoisomerase inhibitors irinotecan and teniposide, the nucleoside analog trifluridine, and bleomycin. Additionally, UVC light, ionizing radiation, and NER deficiency were also strong intracellular H_2_S inducers. Intriguingly, teniposide and UVC treatment increased the cellular sulfane sulfur fraction, indicating that the larger sulfur pool was also altered by genotoxic stress [85]. To further characterize the role of the DDR in these responses, immortalized murine embryonic fibroblasts (MEFs), with and without PARP-1 expression, were treated with UVC or teniposide [85]. PARP-1 loss significantly decreased H_2_S induction by either agent, strongly implicating a role for early DDR events in H_2_S induction [85].

Autophagy is a significant part of the DDR [86]. In MEFs knockdown of ATG5 and ATG7, two autophagy regulators, attenuated LC3I to LC3II conversion (also an autophagy marker) initiated by teniposide. Importantly, this was accompanied by a concomitant decrease in H_2_S induction, linking H_2_S induction, the DDR, and autophagy [85]. ATG5 knockout MEF viability (which showed teniposide hypersensitivity) was partially rescued from teniposide toxicity by concomitant AP39 treatment, indicating that H_2_S plays a cellular protective role in genotoxic exposure [85]. AMPK is activated by PARP-1 and is another DDR mediator [86]. In AMPK, double α1/α2 subunit knockout MEFs, H_2_S and autophagy induction with teniposide were also attenuated, again consistent with a requirement for autophagy in maximal H_2_S induction by genoclastic agents [85].

Lastly, teniposide treatment significantly induced CSE mRNA and protein in MEFs. CSE expression ablation resulted in a small but significant decrease in cellular H_2_S levels with teniposide treatment, indicating a possible role of CSE in the DDR [85]. Under genotoxic stress CSE mRNA is induced by the ATF4 transcription factor [87]. In MEFs, ATF4 expression ablation resulted in both attenuated CSE induction and lowered cellular H_2_S levels following teniposide treatment. Additionally, treatment of wild-type and ATG5 knockout MEFs with PAG decreased teniposide-induced H_2_S in both cell types. These results suggest an additive role for ATF4-mediated CSE expression and autophagy in H_2_S induction upon genotoxic stress [85]. These interactions are summarized in Figure 1.

## 7. Conclusions and Future Directions: Cancer Therapy and Cardiovascular Disease

The role of H_2_S in DNA repair and the DDR is an established, but as yet under researched area with few studies on the subject [11,12,13,85]. Due to the importance of genomic stability maintenance, DNA repair, and H_2_S in normal and pathophysiology, a better understanding of this area will undoubtedly lead to a greater understanding of the molecular pathology underlying many human diseases. For example, increased DNA damage in the vasculature is a major cardiovascular disease risk factor, and H_2_S is a major regulator of cardiovascular functions [14,88]. However, there is relatively little data on the role of the DDR and DNA repair in the cardiovascular system, although intriguingly murine models of defective NER show premature vascular senescence, increased vascular stiffness, and elevated blood pressure [89]. As the ATR kinase regulates NER, there may be an as yet undiscovered role for ATR in vascular diseases [1,2,3,12,89]. Additionally, the DDR protein ATM plays a poorly defined role in cardiovascular disease, with individuals and mice with only one functional ATM gene showing elevated cardiovascular disease, while a high fat diet suppresses ATM protein expression in wild-type animals [90,91]. A role for ATM in H_2_S metabolism has not yet been identified.

Increased H_2_S synthesis and CBS, CSE, and/or 3-MST expression promote cancer progression in several human malignancies, and H_2_S synthesis inhibitors have been proposed as a cancer treatment [5,11,12,17,69,92,93]. In addition, ATR inhibitors are showing promise in phase I and II clinical trials for lung, ovarian, cervical, urothelial, and advanced sold tumors [93]. Since ATR inhibition lowers cellular H_2_S concentrations, this may be an unexamined aspect of ATR inhibition in cancer therapeutics [12,92,93]. Similarly, lowered cellular H_2_S levels with ATR inhibition could, in turn, lower MEK1 and EXOG sulfhydration and activation, attenuating mitochondrial DNA repair and nuclear DNA repair mediated by the MEK1–ERK1/2–PARP-1 axis [11,12,13]. Interestingly, ATR inhibition increases the effectiveness of PARP inhibitors in cancer therapy, supporting this hypothesis [94,95,96]. Thus, H_2_S may form a molecular link uniting different aspects of DNA repair (summarized in Figure 1).

Previously, we reviewed the role of H_2_S in DNA repair with an emphasis on ATR function [69]. Here, we extended this review, as current data strongly implicate a role for other DDR proteins, especially ATM in H_2_S regulation [1,2,3,62,63,64]. Additionally, the data by Jiang et al. demonstrate that H_2_S, autophagy, and the DDR are intricately interconnected, further highlighting the function of H_2_S in the most basic functions regulating cell survival [85]. Support for this also comes from the known roles of the DDR proteins in regulating autophagy [85,97].

ATR, ATM, and DNA-PK are central to the DRR and DNA repair [1,2,3,95,96]. All three proteins share extensive sequence and substrate overlap, and synthetic lethal relationships exist between them [1,2,3,96]. Moreover, all three proteins regulate autophagy and mitochondrial function and viability [97,98,99,100]. These observations, combined with the ancient and parallel origins of H_2_S biochemistry and DNA repair, and with H_2_S now linked to mitochondrial and nuclear DNA repair, suggest that ATM and DNA-PK may also regulate aspects of H_2_S metabolism [1,2,3,11,12,13,17,52,53,54,55,56,57,58]. Although only hypotheses, these ideas could be easily tested. In summary, the role of H_2_S in the regulation of the DDR and DNA repair is a new and exciting area of inquiry and should give useful and profound insights into normal and pathophysiology.

## Figures and Tables

**Figure 1 antioxidants-10-01820-f001:**
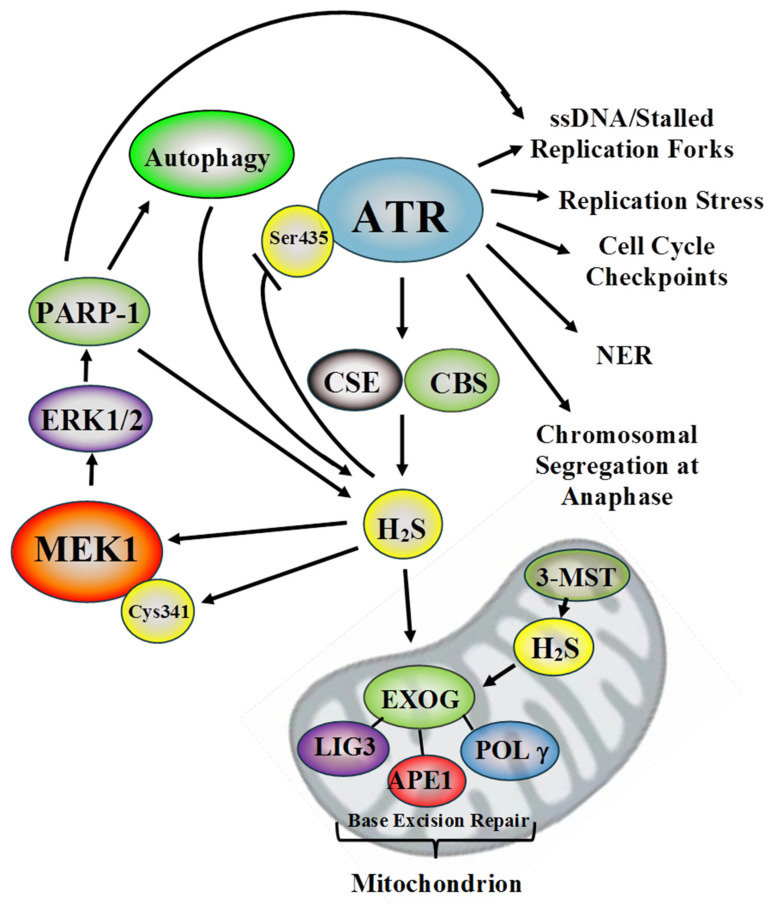
A summary of known pathways by which H_2_S regulates mitochondrial and nuclear DNA repair integrated into one model [11,12,13,85].

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
