# Peer review of "Bad Smells and Broken DNA: A Tale of Sulfur-Nucleic Acid Cooperation"

_antioxidants, 2021, doi:10.3390/antiox10111820_

Round 1
Reviewer 1 Report
In this manuscript, Shackelford et al. summarize the recent development about the role of hydrogen sulfide in DDR and DNA repair. It includes a general description of H2S and the rationales for its participation in DDR and DNA repair, as well as a summary of the recent findings of H2S in DDR and DNA repair. In general, the authors addresses a contemporary topic of interest to general readers. The manuscript is well written and organized.
The manuscript is partially weakened by the fact that the authors have published a review paper entitled hydrogen sulfide and DNA repair in redox biology (38, 2021). There are some overlaps in content, and as a review, it should be acceptable. However, it would be nice to mention the published paper and briefly describe the differences between the two documents. It appears to this review this manuscript is better outlined and organized.
Minor points
Several minor grammatical errors should be corrected, such as a repeating "is not" in line 55 and the long sentence in lines 342~244.
Author Response
In this manuscript, Shackelford et al. summarize the recent development about the role of hydrogen sulfide in DDR and DNA repair. It includes a general description of H2S and the rationales for its participation in DDR and DNA repair, as well as a summary of the recent findings of H2S in DDR and DNA repair. In general, the authors addresses a contemporary topic of interest to general readers. The manuscript is well written and organized.
The manuscript is partially weakened by the fact that the authors have published a review paper entitled hydrogen sulfide and DNA repair in redox biology (38, 2021). There are some overlaps in content, and as a review, it should be acceptable. However, it would be nice to mention the published paper and briefly describe the differences between the two documents. It appears to this review this manuscript is better outlined and organized.
- The previous review article was mentioned in this review’s conclusion.
Minor points
Several minor grammatical errors should be corrected, such as a repeating "is not" in line 55 and the long sentence in lines 342~344.
- The long sentence at 342-344 was shortened.
- “is not” was corrected, several typos were removed, and some repeated references were also corrected.
- A new section was added to the review, as a new paper came out on the DDR and H2S This was added as it was requested by a reviewer and it also significantly added to the review.
- The figure was changed to incorporate new data.
Reviewer 2 Report
In the review article titled "Bad Smells and Broken DNA; A Tale of Sulfur-Nucleic Acid Cooperation" by Shackelford, et al., the authors detail in great length and clarity the biology and evolutionary history of hydrogen sulfide as it related to cellular and organismal physiology. They begin with a brief overview of the enzymatic and non-enzymatic production of the gas, how life evolved with it, and how it ultimately plays roles in health and disease. For the major portion of the review article, the authors detail the literature of DNA damage and stability with H2S playing a role in both.
Overall, the article is well written and clear. I have only 2 minor comments to help improve the current work:
1) Line 56: Grammatical error with "is not" repeated.
2) The new article in Cell Chemical Biology titled "Intracellular H2S production is an autophagy-dependent adaptive response to DNA damage" by Jiang, et al (PMID: 34166610) is not included in this review. While this is understandable due to the article having just come out in June, it would benefit the current review to touch upon some of the findings in that paper, as well as integrate their findings into the current model figure for how DNA damage induced-PARP-1 activity also acts upstream of CSE/H2S.
Author Response
In the review article titled "Bad Smells and Broken DNA; A Tale of Sulfur-Nucleic Acid Cooperation" by Shackelford, et al., the authors detail in great length and clarity the biology and evolutionary history of hydrogen sulfide as it related to cellular and organismal physiology. They begin with a brief overview of the enzymatic and non-enzymatic production of the gas, how life evolved with it, and how it ultimately plays roles in health and disease. For the major portion of the review article, the authors detail the literature of DNA damage and stability with H2S playing a role in both.
Overall, the article is well written and clear. I have only 2 minor comments to help improve the current work:
1) Line 56: Grammatical error with "is not" repeated.
- This was corrected, several typos were removed, and some repeated references were also corrected.
- The figure was changed to incorporate new data.
2) The new article in Cell Chemical Biology titled "Intracellular H2S production is an autophagy-dependent adaptive response to DNA damage" by Jiang, et al (PMID: 34166610) is not included in this review. While this is understandable due to the article having just come out in June, it would benefit the current review to touch upon some of the findings in that paper, as well as integrate their findings into the current model figure for how DNA damage induced-PARP-1 activity also acts upstream of CSE/H2S.
- This article was incorporated into the manuscript in a new section. It significantly added to and improved the review.